# Diagnostic and Therapeutic Pathways of Intramuscular Myxoma

**DOI:** 10.3390/diagnostics12071573

**Published:** 2022-06-28

**Authors:** Alonja Reiter, Katharina Trumm, Tobias M. Ballhause, Sebastian Weiss, Karl-Heinz Frosch, Alexander Korthaus, Ulrich Bechler, Anna Duprée, Andreas Luebke, Peter Bannas, Carsten W. Schlickewei, Matthias H. Priemel

**Affiliations:** 1Department of Traumatology and Orthopedics, University Medical Center Hamburg-Eppendorf, 20251 Hamburg, Germany; katharina.trumm@stud.uke.uni-hamburg.de (K.T.); t.ballhause@uke.de (T.M.B.); s.weiss@uke.de (S.W.); k.frosch@uke.de (K.-H.F.); a.korthaus@uke.de (A.K.); u.bechler@uke.de (U.B.); c.schlickewei@uke.de (C.W.S.); priemel@uke.de (M.H.P.); 2Department of Trauma Surgery, Orthopedics and Sportstraumatology, BG Hospital Hamburg, 21033 Hamburg, Germany; 3Department of General, Visceral and Thoracic Surgery, University Medical Center Hamburg-Eppendorf, 20251 Hamburg, Germany; a.dupree@uke.de; 4Department of Pathology, University Medical Center Hamburg-Eppendorf, 20251 Hamburg, Germany; luebke@uke.de; 5Department of Diagnostic and Interventional Radiology and Nuclear Medicine, University Medical Center Hamburg-Eppendorf, 20251 Hamburg, Germany; p.bannas@uke.de

**Keywords:** intramuscular myxoma (IMM), tumor imaging, benign tumor, soft tissue tumor

## Abstract

Intramuscular myxomas (IMMs) are benign tumors. Evidence regarding diagnostic and therapeutic pathways is rare, and guidelines do not exist due to their low incidence. The aim of this study was a retrospective analysis at a university cancer center and the interdisciplinary re-evaluation of the individual diagnostic and therapeutic procedures. Overall, 38 patients were included in the study. IMMs occurred mostly in middle-aged women. At the time of first consultation, 57.9% had few symptoms or were asymptomatic. In 92.1% of the cases, the tumor was localized in the extremities. The lower extremity was affected in 73.7%. The average size of IMMs was 5.0 cm. The proximally located tumors in the gluteus, thighs, and upper arms were significantly larger (*p* = 0.02) than the distally-located tumors in the forearms and lower legs. An MRI was performed in 97.4%. Based on imaging, an IMM was suspected in 5.6% by radiologists and in 54.1% by musculoskeletal surgeons. An incision biopsy was performed in 68.4% and led in 100.0% to the right histopathological diagnosis. In total, 89.5% of IMMs were resected. Postoperative complications requiring revision occurred in 8.8%. Recurrences or degenerations of IMMs were not reported in any of these cases.

## 1. Introduction

Intramuscular myxomas (IMMs) are rare benign soft tissue tumors, first described in 1965 by Enzinger [1]. In most cases, they are associated with very slow growth over several years [2]. Patients often present with mild symptoms, such as a painless palpable mass or pressure pain [3]. In some cases, patients are asymptomatic, and the IMM is an incidental diagnosis.

The incidence of IMMs is reported to be 0.11 per 100,000 [4]. In comparison, the incidence of lipomas is 100 per 100,000 [5]. Due to this low incidence, in the literature, only a few retrospective studies and some single case reports are available [6,7,8,9,10,11]. Current evidence is missing. Diagnostic and therapeutic guidelines for IMMs do not exist. In most cases, unclear soft tissue tumors are diagnosed and treated according to the sarcoma guidelines [12].

The aim of this study was the single center analysis of our patient collective at a university cancer center and the interdisciplinary surgical, radiological, and pathological re-evaluation of the individual diagnostic and therapeutic procedures and the development of a diagnostic and therapeutic pathway. 

## 2. Methods

The study was approved by the institutional ethics committee of the medical association Hamburg, Germany (WF-066/21).

All patients with a histopathological diagnosis of IMM at our university medical center were included in the anonymized retrospective analysis. The comorbidities, noxae, and demographic data were collected from the patient’s electronic medical chart. In addition to the findings recorded at our medical center, radiological findings were also included in the analysis. The imaging was performed by external radiologists, and all were fellowship-trained. All patients were referred to our university cancer center after imaging. A magnet resonance tomography (MRI) was performed in 37 cases, with only one without the application of contrast medium (CM). Due to contraindications, a computed tomography (CT) with CM had to be performed in one case instead of an MRI. At our cancer center, all MR-images and CT-images were reviewed by a musculoskeletal surgeon with more than 10 years of experience. 

In total, this study evaluated data of 38 patients with an IMM between 09/2011 and 12/2021. Overall, 25 female and 13 male patients with a histopathological diagnosis of an IMM were included in the analysis. The average age was 53.1 years (range: 30.0–79.0; SD ± 13.7). The BMI was on average 25.9 kg/m^2^ (range: 19.7–39.3; SD ± 4.4).

In all cases, the histopathological diagnosis was confirmed in our department of pathology. 

The data were analyzed with GraphPad Prism V 9 (Ja Lolla, CA, USA) by the means of descriptive statistics. 

## 3. Results

IMMs occurred mostly in middle-aged women (mean: 50.3 years; SD ± 11.1). In addition, our patients showed a tendency of being overweight, with a BMI of 25.9 kg/m^2^. In 30 of 38 patients (78.9%), at least one chronic disease was reported. Most frequently, other tumors were reported. In 8 cases (21.1%) malignant tumors (e.g., prostate or breast carcinoma) and in 5 cases (13.2%) benign tumors (e.g., myoma, lipoma) were known. Thirteen patients (34.2%) had a previous history of cardiac disease (e.g., arterial hypertension). In addition, hypothyroidism was known in 11 cases (29.0%).

In 35 cases (92.1%), the tumor was localized in the extremities. In 28 cases (73.7%), the lower extremity was affected, and in 7 cases (18.4%), the upper extremity was affected, but only in one case was below the elbow affected. In 3 cases (7.9%), the tumor was localized in the region of the trunk. The left side of the body was affected in 26 cases (68.4%). 

At the time of first consultation, 22 patients (57.9%) had few symptoms or were asymptomatic. Overall, 23 patients (60.5%) complained of only a painless palpable mass at the beginning. Still, 5 of these 23 patients (21.7%), reported pain in the tumor or the surrounding area over time. Additionally, pressure- and movement-induced pain was reported in 11 cases (29.0%) as one of the first symptoms. However, 4 patients (10.5%) were completely asymptomatic. In these cases, the IMM was an incidental finding.

MRI was performed in 37 of 38 cases (97.4%). Due to claustrophobia, a CT with CM had to be performed in one case instead of an MRI. An MRI with intravenous CM was performed in 36 cases (94.7%). However, the contraindication to CM administration in one case was not reported. 

On MRI, IMMs presented as a homogeneous mass with an isointense signal on T1-weighted and a hyperintense signal on T2-weighted sequences, with a heterogeneous CM enhancement (Figure 1a–c). The average size of IMMs on MRI and CT was 5.0 cm (range: 2.0–9.5; SD ± 2.1) in the plane of the largest diameter. The proximally-located tumors in the gluteus, thighs, and upper arms were significantly larger (*p* = 0.02), with an average length of 5.2 cm (range: 2.0–9.5; SD ± 2.0) compared to the distally-located tumors in the forearms and lower legs, with an average size of 3.2 cm (range: 2.3–4.0; SD ± 0.6). IMMs in the trunk area were the largest, with an average of 7 cm (range: 5.3–8.6; SD ± 1.7). The volume of IMMs showed a wide range between 4.2 and 637.7 cm^3^, with a mean value of 86 cm^3^ (SD ± 114.2). 

A radiological provisional diagnosis based on MRI or CT images was made by the radiologist in 36 cases (94.7%). The radiologists suspected malignant soft tissue tumors in 15 cases (41.7%). In 9 cases (25.0%), an unclear soft tissue tumor was described without naming a dignity or an origin. In 12 cases (33.3%), a benign tumor was suspected. However, an IMM was named as a likely diagnosis in only two cases (5.6%) (Figure 2a).

At our university cancer center, all MRI and CT-images were viewed by experienced musculoskeletal surgeons, independent of the radiologist’s assessment of the tumor. The surgeons named their own suspected diagnosis on the basis of the images, anamnesis, and physical examination. An IMM was named here as the provisional diagnosis in 20 cases (54.1%). Furthermore, in 17 cases (46.0%), a schwannoma was considered, thus becoming the most important differential diagnosis. In 6 cases (16.2%), a benign neoplasm other than IMM was considered, and in 11 cases (29.7%), a type of soft tissue sarcoma was suspected (Figure 2b).

Prior to tumor resection, a histopathological confirmation of diagnosis was secured in 26 cases (68.4%) by biopsy. In one case, a fine needle aspiration biopsy was performed outside of our center. In this case, the available material was not sufficient for a reliable diagnosis, so a second biopsy (an incisional biopsy) was performed.

In all cases (100.0%) in which an incision biopsy was performed, the diagnosis could successfully be confirmed histopathologically. The mean time to histopathological diagnosis through biopsy or resection was 12.1 days (range: 2.0–53.0; SD ± 10.5) after the first medical consultation.

After the histopathological diagnosis, a complete resection of the IMMs was performed in 22 of 26 cases (84.6%). Four patients (15.4%) with a histopathologically-assured diagnosis of an IMM decided against a tumor resection after exclusion of a malignant tumor by biopsy due to mild symptoms. There were no signs of degeneration in any IMM in histopathologic analysis.

In 12 cases (31.6%), primary complete tumor resections were performed without prior histopathological confirmation of the diagnosis. The most frequent reasons for a primary resection were a liquid myxoid tumor appearance and the intention to avoid local contamination and the high-grade clinical and radiological suspicion of a benign mass or deep-lying or small tumors (<3 cm). In 5 of these cases (41.6%), the tumor was removed with a safety margin (in-sano-resection). In 7 cases, a marginal tumor resection with excision of the tumor and surrounding capsule was performed, because of the benign tumor appearance on MRI. 

Postoperative complications requiring revision occurred in three cases (8.8%). In two cases, wound infections and wound healing disturbances occurred. In one case, direct wound closure was not possible due to the large size of the IMM, so a split skin grafting was performed in a further surgery. 

A regular postoperative tumor follow-up was not performed in any of the cases after histopathological exclusion of a malignant tumor. 

To date, there have been no further consultations due to a recurrence of the tumor. 

## 4. Discussion

Our analysis revealed that IMMs occur more frequently in middle-aged women. In our patient collective, IMMs affected, in most cases, the extremities, whereas the lower extremity was predestined. This is in common with other studies. Only in rare cases do IMMs occur on the trunk or other parts of the body [13,14]. Furthermore, our patient population showed a tendency towards being overweight. Other studies reported, as well, a higher prevalence of soft tissue tumors by increased mass of deep soft tissue [15].

Similar to other slow-growing soft tissue tumors, most patients of our study presented with mild or even no symptoms on initial consultation. This is in line with available case series. In many cases, pain and impairment only become apparent in later stages as the size of the tumor increases and neighboring structures are disturbed and irritated [11,16].

An MRI with CM was performed in all cases without contraindication for MRI or CM administration according to the guidelines for sarcoma and soft tissue tumors [12]. The radiological findings on MRI in the different cases were very similar in comparison. Overall, IMMs in our study were described as a sharply demarcated mass with a surrounding edema and cystic and/or liquid parts. This is in line with the available literature [17,18]. In T1-weighted sequences, IMMs were described as hypointense and, in T2-weighted sequences, as hyperintense with an inhomogeneous enhancement after CM administration. On CT, IMMs were described as sharply demarcated, as well, with central hypodense and peripheral hyperdense lesions and inhomogeneous internal structures. On ultrasonography, IMMs presented as an echo-inhomogeneous and solid tumor with cystic and gelatinous parts surrounded by a solid capsule. These descriptions of IMMs are comparable to the existing literature [16,19].

The determined size of IMMs in our series ranged between 2.0 and 9.5 cm. Other studies reported a similar size, with diameters from 4.0 to 5.9 cm. However, a tumor size larger than 10 cm is unusual and was only reported in a few cases [1,7,8,9,11,16,19]. 

The radiologically-suspected diagnoses were heterogeneous. The radiologists described both malignant or benign tumors and benign non-tumor lesions and often did not commit to one diagnosis. IMMs were suspected only in two cases. More frequently, soft tissue sarcomas and hematomas were considered. These results are reflected in the literature, describing a high rate of misinterpretation of tumors on the basis of MRI images [20]. 

In contrast, IMMs were the most-frequently-suspected tumor at the university medical center by musculoskeletal surgeons. However, due to the comparable clinical and radiological appearance, schwannomas were the most important differential diagnosis (Figure 1d–f). In some cases, soft tissue sarcomas were considered too. These results are following other studies, which also reported inaccurate diagnoses like schwannoma, liposarcoma, fibroma, and myxofibrosarcoma [1,4,8].

Nevertheless, the question remains of why radiologists considered so rarely an IMM as a differential diagnosis. On the one hand, this is certainly explained by the low incidence of IMMs. On the other hand, the imaging was performed in an outpatient setting, so in many cases, the assessing radiologists had probably little to no experience with IMMs compared to musculoskeletal surgeons at a specialized tumor center. This observation demonstrates the importance of a reevaluation of diagnostics by a specialized musculoskeletal surgeon. This results in a better outcome at a specialized center with experience regarding rare tumor diseases [21,22,23].

The question remains, why did both radiologists and musculoskeletal surgeons at the tumor center mostly not commit to only one diagnosis and often considered soft tissue sarcoma, schwannoma, and hematoma too? 

Although it is feasible to differentiate IMMs from other soft tissue tumors based on imaging and physical examination, a final differentiation is difficult and requires accurate knowledge [24]. 

Following other studies [1,2,8,25,26], we considered two differential diagnoses in particular as important clinical and radiological differential diagnoses: schwannoma and myxoid liposarcoma (Table 1). 

Schwannomas are benign nerve sheath tumors that descend from Schwann cells [27]. They appear equally often in women and men and occur in most cases in adults between 20 and 50 years. Usually, they present as a painless mass like IMMs. On MRI, they present as a spindle-like shaped mass with a hypo- to isointense signal on T1-weighted and hyperintense signal on T2-weighted sequences, as well as a homogeneous to heterogeneous enhancement after CM administration [24,28].

Myxoid liposarcoma are the most prevalent type of myxoid sarcomas and are often mistaken for IMMs [1,25]. They are equally common in men and women, with a peak age of 40 to 50 years, and they often present as a large, painless, intermuscular located mass mostly in the lower extremities. On MRI they appear (similar to IMMs) hypointense on T1- and hyperintense on T2-weighted sequences, with the difference of hyperintense foci on T1-weighted sequences, presenting adipose components within the mass [24].

Histopathological confirmation of an IMM was achieved in all cases where an incision biopsy was performed. Additionally, in all cases with subsequent tumor resection, this diagnosis was confirmed. Although performed in only one case, our analysis demonstrates the potential for error in fine needle aspiration biopsies. This is in line with the recommendations of sarcoma guidelines and previously published studies, declaring that a fine needle aspiration biopsy is not an evidence-based method to obtain high-quality biopsy samples [29,30]. Although not performed in our study, a core-needle biopsy can also be discussed as an alternative to incision biopsies [31].

However, a histopathological diagnosis should be performed promptly in every case [32]. Even in cases with a radiological suspicion of an IMM, the histological diagnosis confirmation should be accelerated. As our evaluation demonstrated, in many cases, IMMs resemble soft tissue sarcomas and cannot always be differentiated, even by experienced musculoskeletal surgeons [33]. 

In our study, in-sano-resections of the IMMs were performed much more frequently in patients with a histopathologically-unconfirmed diagnosis (41.6%) compared to patients with a histopathologically-confirmed diagnosis (18.2%). However, if a biopsy is not possible for individual reasons, e.g., due to a high risk of local tissue contamination, a superficial small (<3 cm), or a very deep tumor, an in-sano-resection should always be performed. This avoids tumor tissue remaining in the circumstance of a postoperative confirmed soft tissue sarcoma. In cases with a histopathological verification of an IMM by biopsy before resection, a tissue-sparing marginal tumor resection can be performed. However, to avoid recurrences, the tumor-surrounding capsule should be completely removed in every case. 

To date, no recurrences of an IMM after a complete resection were reported, neither in any previously published case report, nor in any patient in our study [34]. Based on the current literature and our analysis, there are no indications regarding the potential of degeneracy of IMMs [35,36]. Therefore, we do not consider standardized radiological follow-up examinations to be necessary.

**Table 1 diagnostics-12-01573-t001:** Radiological characteristics of IMMs and its most common differential diagnoses.

Diagnosis	Description	Ultrasonography	Computed Tomography	Magnetic Resonance Imaging
Intramuscular Myxoma [24,37,38,39]	*benign, myxoid soft tissue tumor of mesenchymal origin*	*hypoechoic mass with posterior acoustic enhancement*	*homogeneous mass;* *attenuation lower than muscle and higher than water;* *inhomogeneous, peripheral CM enhancement*	*homogeneous mass with hypo- to isointense signal on T1w;* *hyperintense signal on T2w;* *heterogeneous CM enhancement*
Schwannoma [24,27,28,40]	*benign nerve sheath tumor descending from Schwann cells*		*low attenuated mass;* *heterogeneous CM enhancement*	*hypo- to isointense signal on T1w;* *hyperintense signal on T2w;* *homogeneous or heterogeneous CM enhancement*
Myxoid Liposarcoma [24,41]	*malignant, myxoid soft tissue tumor descending from primitive mesenchymal cells*		*well-defined, lobulated mass; mainly low attenuation*	*hypointense signal on T1w with some hyperintense foci;* *hyperintense signal on T2w;* *homogeneous, heterogeneous, or no CM enhancement*

If an IMM is confirmed by biopsy, the necessity for a resection can be discussed on an individual patient basis. There is no general recommendation for resection in asymptomatic patients. A clinical or radiological follow-up after resection in cases with a histopathologically-confirmed IMM is not required. However, clinical follow-up should be considered in cases with no or an incomplete tumor resection or postoperative complications.

In summary of the results of our study and the existing literature, we recommend the diagnostic and therapeutic pathway shown in Figure 3. In cases with a slowly growing painless mass located in the extremities, MRI should be performed as the next diagnostic step according to the current literature and guidelines for sarcomas. An MRI provides the ability to assess prior to biopsy whether the tumor is more likely to be benign or malignant, thus enabling a critical planning of the biopsy. If the findings on MRI are classic for IMMs (Table 1), an IMM can be suspected at a high grade. In these cases, a decision between a direct resection and a prior biopsy must be made. If a biopsy is possible in consideration of the tumor location, tumor size, and tumor consistency, then it should be performed. In this occasion, we recommend an incisional biopsy in order to be able to secure a sufficient amount of tumor for histopathological analysis. A fine needle aspiration biopsy should not be performed, as both our study and the existing literature show that this form of biopsy is vulnerable to error due to the small amount of tumor material. If the biopsy confirms an IMM, a marginal resection can be performed subsequently. However, if a biopsy is not possible, we recommend a complete tumor resection with a safety margin. This avoids the necessity of a subsequent resection in the unlikely event of a malignant tumor, and potential contaminations can be avoided. 

This study has some limitations. First, the study design is retrospective. Second, in all cases individual diagnostic and therapeutic pathways were chosen, causing both a performance and a selection bias. Third, IMMs are very rare tumors, resulting in a small patient population and reduced available literature. Fourth, standardized follow-up examinations were not performed in our study, so information regarding tumor recurrences are based on passive information collection.

## 5. Conclusions

In synopsis of the individual diagnostic and therapeutic pathways and the existing literature, we would recommend the pathway shown in Figure 3. Even in cases with high suspicion of an IMM, the diagnosis should be confirmed by biopsy, as recommended in the current guidelines for sarcomas. If there is a higher perioperative risk profile for local contamination, a direct excision can be discussed. However, in these cases, an in-sano-resection should be aimed for. 

## Figures and Tables

**Figure 1 diagnostics-12-01573-f001:**
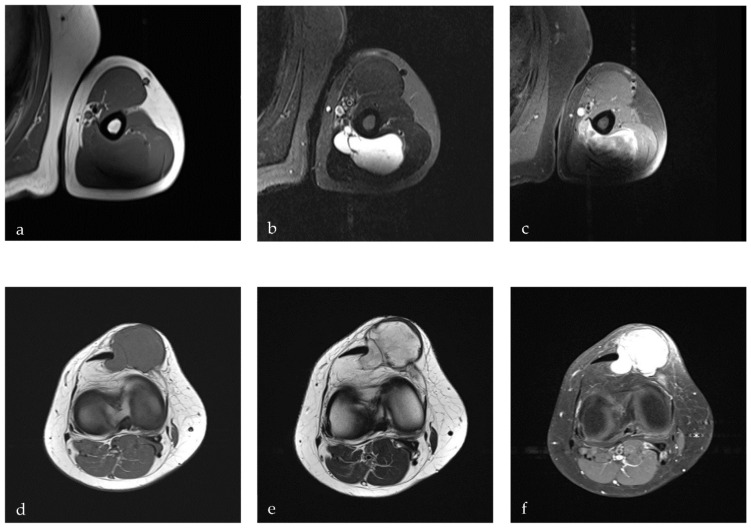
IMM and schwannoma appearance on MRI: (**a**) IMM on T1w sequences; (**b**) IMM on T2w sequences; (**c**) IMM after CM administration; (**d**) schwannoma on T1w sequences; (**e**) schwannoma on T2w sequences; (**f**) schwannoma after CM administration with CM.

**Figure 2 diagnostics-12-01573-f002:**
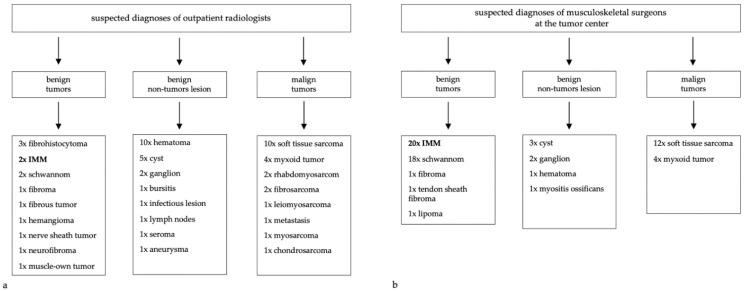
Suspected diagnoses by radiologists and musculoskeletal surgeons (multiple diagnosis per patient possible): (**a**) reported suspected diagnoses by the outpatient radiologists; (**b**) reported suspected diagnoses by the musculoskeletal surgeons at the specialized tumor center.

**Figure 3 diagnostics-12-01573-f003:**
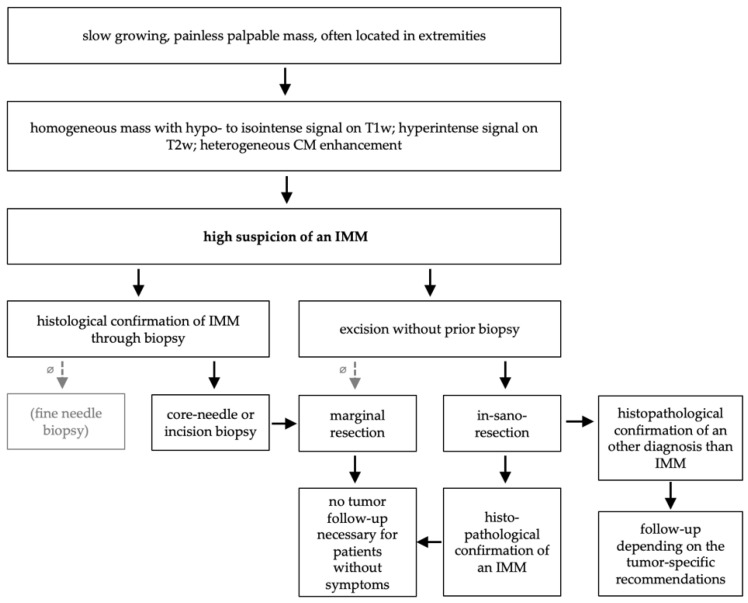
Recommended diagnostic and therapeutic pathway for IMM at a specialized tumor center. Based on our analysis and the current literature.

## Data Availability

The raw data are available upon reasonable request from the corresponding author.

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
