# Peer review of "Diagnostic and Therapeutic Pathways of Intramuscular Myxoma"

_diagnostics, 2022, doi:10.3390/diagnostics12071573_

Round 1
Reviewer 1 Report
authors should be congratulated for their manuscript.
Author Response
Dear Reviewer,
Thank you for your time and effort in reviewing our manuscript. We are very pleased with your positive feedback.
Best regards.
Reviewer 2 Report
This manuscript is a retrospective cohort analysis of a small number of patients with benign soft tissue tumors. IMM seems to pose no true health risk, is relatively symptomless unless large, and extremely rare and consequently is not well diagnosed outside of specialist institutes. The authors have looked at various parameters to derive an algorithm to identify these tumors and recommend treatment.
Some comments that should be considered are below.
1. It seems that incision histopathology is fool proof and should be recommended over imaging for palpable mostly symptomless masses in muscle. This would also be cost effective over expensive imaging studies. Would the authors comment on their choice of imaging as the first approach in figure 3.
2. The authors state that this was approved by the institutional ethics committee. Was there a need for informed consent? If so, this should be clearly stated.
3. In results lines 85-87. The number of MRIs do not add up.
4. In discussion lines 154-157 might really belong in results.
Author Response
Dear Reviewer,
Thank you for the detailed review and the helpful comments you made to improve the manuscript.
To 1.
Thank you for this very important point.
The question of whether and what imaging should be done for tumors is also frequent for us in our daily routine.
Fig. 3 is not only based on our data evaluation but also includes results and recommendations of other studies and the guideline for sarcomas. Here, MRI is recommended explicitly in cases of unclear soft tissue tumors. Although the biopsy provided definite histopathological results in all our cases, it must be considered that other tumors - in the worst-case malignant tumors with possibly liquid parts - have a similar clinical presentation. Performing a biopsy in these cases would risk contamination of the surrounding tissue.
We have added this in lines 255-258.
To 2.
Thank you for this important question.
Informed consent was not required in these cases because it was an anonymized retrospective evaluation. However, even in these cases, we require approval from our regional ethics committee outlining the use of the analyzed data.
To 3.
Thank you for this comment.
The paragraph certainly leads to misunderstandings. We have corrected the paragraph as follows (lines 87-90):
"MRI was performed in 37 of 38 cases (97.4%). Due to claustrophobia, a CT with CM had to be performed in one case instead of an MRI. An MRI with intravenous CM was performed in 36 cases (94.7%). However, the contraindication to CM administration in one case was not reported."
To 4.
Thank you for this good point.
We have added the BMI to the results (lines 70-71):
"In addition, our patients showed a tendency of overweight with a BMI of 25.9 kg/m2."Additionally, this part was changed in the discussion as follows:
"Furthermore, our patient population showed a tendency towards overweight. Other studies reported as well a higher prevalence of soft tissue tumors by increased mass of deep soft tissue [15]."
Best regards.
Reviewer 3 Report
This reviewer would like to congratulate the authors on their work. The study has some limitations that the authors acknowledge and discuss, but it is nevertheless intriguing enough to be published in its present form.
Author Response
Dear Reviewer,
Thank you for your support in the publication process. You are right, the study has some limitations. However, we are pleased that you agree to accept the manuscript despite the limitations.
Best regards.